# Development of standard indicators to assess use of electronic health record systems implemented in low-and medium-income countries

Philomena Ngugi [1,2]*, Ankica Babic[1,3], James Kariuki[4], Xenophon Santas[5], Violet Naanyu[6], Martin C. Were[2,7]

1 Department of Information Science and Media studies, University of Bergen, Bergen, Norway, 2 Institute of Biomedical Informatics, Moi University, Eldoret, Kenya, 3 Department of Biomedical Engineering, Linköping University, Linköping, Sweden, 4 Division of Global HIV & TB, Center for Global Health, Centers for Disease Control and Prevention, Atlanta, Georgia, United States of America, 5 Center for Global Health, Centers for Disease Control and Prevention, Atlanta, Georgia, United States of America, 6 School of Arts and Social Sciences, Moi University, Eldoret, Kenya, 7 Vanderbilt Institute of Global Health, Vanderbilt University Medical Center, Nashville, Tennessee, United States of America

* waruharip@gmail.com

**Data Availability Statement:** All relevant data are within the paper and its Supporting information files.

## Abstract

### Background

Electronic Health Record Systems (EHRs) are being rolled out nationally in many low- and middle-income countries (LMICs) yet assessing actual system usage remains a challenge. We employed a nominal group technique (NGT) process to systematically develop high-quality indicators for evaluating actual usage of EHRs in LMICs.

### Methods

An initial set of 14 candidate indicators were developed by the study team adapting the Human Immunodeficiency Virus (HIV) Monitoring, Evaluation, and Reporting indicators format. A multidisciplinary team of 10 experts was convened in a two-day NGT workshop in Kenya to systematically evaluate, rate (using Specific, Measurable, Achievable, Relevant, and Time-Bound (SMART) criteria), prioritize, refine, and identify new indicators. NGT steps included introduction to candidate indicators, silent indicator ranking, round-robin indicator rating, and silent generation of new indicators. 5-point Likert scale was used in rating the candidate indicators against the SMART components.

### Results

Candidate indicators were rated highly on SMART criteria (4.05/5). NGT participants settled on 15 final indicators, categorized as system use (4); data quality (3), system interoperability (3), and reporting (5). Data entry statistics, systems uptime, and EHRs variable concordance indicators were rated highest.

**Funding:** Author: MCW Norwegian Programme for Capacity Development in Higher Education and Research for Development (NORAD: Project QZA-0484) through the HITRAIN program) https://norad.no/en/front/funding/norhed/projects/#&sort=date The funders had no role in study design, data collection and analysis, decision to publish, or preparation of the manuscript.

**Competing interests:** The authors have declared that no competing interests exist.

## Conclusion

This study describes a systematic approach to develop and validate quality indicators for determining EHRs use and provides LMICs with a multidimensional tool for assessing success of EHRs implementations.

## Introduction

Electronic Health Record Systems (EHRs) are increasingly being implemented within low-and middle-income countries (LMICs) settings, with the goal of improving clinical practice, supporting efficient health reporting and improving quality of care provided [1,2]. System implementation is the installation and customization of information systems in organizations making them available for use to support service delivery, e.g. EHRs in healthcare [3,4]. National-level implementations of EHRs in many LMICs primarily aim to support HIV care and treatment, with funding for these systems coming from programs such as the US President's Emergency Plan for AIDS Relief (PEPFAR) [5,6]. Several countries, such as Rwanda, Uganda, Mozambique, and Kenya, have gone beyond isolated and pilot implementations of EHRs to large-scale national rollout of systems within government-run public facilities [7]. For example, Kenya has had over 1000 electronic medical systems (EMRs) implementations progressively since 2012 in both private and public facilities supporting patient data management mainly in HIV care and treatment [8]. With such large-scale EHRs implementations, developing countries are finding themselves in the unenviable position of being unable to easily track the status of each implementation, especially given that most of the EHRs implementations are standalone and are distributed over large geographical areas. A core consideration is the extent to which the EHRs implemented are actually in use to support patient care, program monitoring, and reporting. Without robust evidence of use of the implemented EHRs, it becomes difficult to justify continued financial support of these systems within these resource-constrained settings and to realize the anticipated benefits of these systems.

In LMICs, implementation of EHRs within a clinical setting does not automatically translate to use of the system. While the evidence is mounting on the benefits of EHRs in improving patient care and reporting in these settings, a number of studies reveal critical challenges to realizing these benefits [9–11]. Some of these challenges include: poor infrastructure (lack of stable electricity, unreliable Internet connectivity, inadequate computer equipment), inadequate technical support, limited computer skills and training, and limited funding [12–17]. Additionally, implementation of EHRs is complex and can be highly disruptive to conventional workflows. Disruption caused by the EHRs can affect its acceptance and use; this is more likely to happen if the implementation was not carefully planned and if end-users were not adequately involved during all stages of the implementation [18–21]. The use of the EHRs can also be affected by data quality issues, such as completeness, accuracy, and timeliness [22]. This is a particular risk in LMICs given the lack of adequate infrastructure, human capacity, and EHRs interoperability across healthcare facilities [23].

Although LMICs have embraced national-level EHRs implementations, little evidence exists to systematically evaluate actual success of these implementations, with success largely defined as a measure of effectiveness of the EHRs in supporting care delivery and health system strengthening [24–26]. Success of EHRs implementation depends on numerous factors, and these often go beyond simple consideration of the technology used [19,20]. Many information system (IS) success frameworks and models incorporate a diverse set of success measures, such

as "effectiveness, efficiency, organizational attitudes and commitment, users' satisfaction, patient satisfaction, and system use" [27–34]. Among numerous IS success frameworks and models, "system use" is considered an important measure in evaluating IS success; IS usage being "the utilization of information technology (IT) within users' processes either individually, or within groups or organizations" [29,31]. There are several proposed measures for system use, such as frequency of use, extent of use, and number of system accesses, but these tend to differ between models. The system use measures are either self-reported (subjective) or computer-recorded (objective) [22,29,30].

There is compelling evidence that IS success models need to be carefully specified for a given context [34]. EHRs implementations within LMICs have unique considerations, hence system use measures need to be defined in a way to ensure that they are relevant, meet the EHRs monitoring needs, while not being too burdensome to accurately collect. Carefully developed EHRs use indicators and metrics are needed to regularly monitor the status of the EHRs implementations, in order to identify and rectify challenges to advance effective use. A common set of EHRs indicators and metrics would allow for standardized aggregation of performance of implementations across locations and countries. This is similar to the systems currently in use for monitoring the success of HIV care and treatment through a standard set of HIV Monitoring, Evaluation and Reporting (MER) indicators [35].

All care settings providing HIV care through the PEPFAR program and across all countries are required to report the HIV indicators per the MER indicator definitions. An approach that develops EHRs indicators along the same lines and format as HIV MER indicators assures that the developed EHRs system use indicators are in a format well-familiar to most care settings within LMICs. This approach reduces the learning curve to understanding and applying the developed indicators. In this paper, we present development and validation of a detailed set of EHRs use indicators that follows the HIV MER format, using nominal group technique (NGT) and group validation technique. This was developed for Kenya, however, it is applicable to LMICs and similar contexts.

## Materials and methods

### Identification of candidate set of EHRs use indicators

Using desk review, literature review, and discussions with subject matter experts, the study team (PN, MW, JK, XS, AB) identified an initial set of 14 candidate indicators for EHRs use [36–39] The candidate set of indicators were structured around four main thematic areas, namely: system use, data quality, interoperability, and reporting. System use and data quality dimensions broadly reflect IS system use aspects contained in the DeLone and McLean IS success model, while interoperability and reporting dimensions enhance system availability and use [39]. The focus was to come up with practical indicators that were **s**pecific, **m**easurable, **a**chievable, **r**elevant, and **t**ime-bound (SMART) [40]. This would allow the developed indicators to be collected easily, reliably, accurately, and in a timely fashion within the resource constraints of clinical settings where the information systems are implemented.

Each of the 14 candidate indicators was developed to clearly outline the description of the indicator, the data elements constituting the numerator and denominator, how the indicator data should be collected, and what data sources would be used for the indicator. These details for the indicators were developed using a template adapted from the HIV MER 2.0 indicator reference guide, given that information systems users in most of these implementation settings were already familiar with this template (S1 Appendix) [35]. Nevertheless, it will require short training time for those unfamiliar due the simplicity of the format.

## Nominal group technique (NGT)

NGT is a ranking method that enables a controlled group of nine or ten subject matter experts to generate and prioritize a large number of issues within a structure that gives the participants an equal voice [41]. The NGT involves several steps, namely: 1) silent, written generation of responses to a specific question, 2) round-robin recording of ideas, 3) serial discussion for clarification and, 4) voting on item importance. It allows for equal participation of members, and generates data that is quantitative, objective, and prioritized [42,43]. Nominal group technique (NGT) was used in the study to reach consensus on the final set of indicators for monitoring EHRs use.

## NGT participants

Indicator development requires consultation with broad-range of subject matter experts with knowledge of the development, implementation, and use of EHRs. With guidance from Kenya Ministry of Health (MoH), a heterogeneous group of 10 experts was invited for a two-day workshop led by two of the researchers (M.W. and P.N.) and a qualitative researcher (V.N.). Inclusion in the NGT team was based on the ability of the NGT participant to inform the conversation around EHRs usage metrics and indicators, with an emphasis on assuring that multiple perspectives were represented in the deliberations. The NGT participants' average age was 40 years where majority were males (69%). The participants included: the researchers acting as facilitators; one qualitative researcher (an associate professor and lecturer); two MoH representatives from the Division of Health Informatics and M&E (health information systems management experts); one Service Development Partners (SDPs) representative (oversees EHRs implementations and training of users); four users of the EHRs (clinical officers (2) & Health records information officers (2)); CDC funding agency representative (an informatics service fellow in the Health Information Systems); and two representatives from the EHRs development and implementing partners (Palladium and International Training and Education Center for Health (I-TECH)), who have been involved in the EHRs implementations and who selected sites for EHRs implementations [44,45]. The study participants were consenting adults, and participation in the group discussion was voluntary. All participants filled a printed consent form before taking part in the study. Discussions were conducted in English, with which all participants were conversant. For analysis and reporting purposes, demographic data and roles of participants were collected, but no personal identifiers were captured. The study was approved by the Institutional Review and Ethics Committee at Moi University, Eldoret (MU/MTRH-IREC approval Number FAN:0003348).

## NGT process

The NGT exercise was conducted on April 8–9, 2019, in Naivasha, Kenya. After providing informed consent, the NGT participants were informed about the purpose of the session through a central theme question: "How can we determine the actual use of EHRs implemented in our healthcare facilities?" Participants were first given an overview on the NGT methodology and how it has been used in the past. Given that candidate indicators had already been defined in a separate process, we did not include the first stage of silent generation of ideas. Ten NGT participants (excluding research team members) evaluated the candidate indicators on quality using the SMART criteria on a 5-point Likert scale rating on each of the five quality components. The NGT exercise was conducted using the following five specific steps:

**Step 1: Clarification of indicators**. For each of the 14 candidate indicators, the facilitator took five minutes to introduce and clarify details of the candidate indicator to ensure all

participants understood what each indicator was meant to measure and how it would be generated. Where needed, participants asked questions and facilitators provided clarifications.

**Step 2: Silent indicator rating**. The participants were given 10 minutes per indicator and were asked to: (1) individually and anonymously rate each candidate indicator on each of the SMART dimensions using a 5-point Likert scale for each dimension where 1 = Very Low, 2 = Low, 3 = Neutral, 4 = High, and 5 = Very high level of quality; (2) provide an overall rating of each indicator on a scale from 1–10, with 10 being the highest overall rating for an indicator; (3) indicate whether the indicator should be included in the final list of indicators or removed from consideration; and (4) provide written comments on any aspect regarding the indicator and their rating process. To help with this process, a printed standardized indicator ranking form was provided (S2 Appendix), and the indicator details were projected on a screen.

**Step 3: Round-robin recording of indicator rating**. Each participant in turn was asked to give their overall rating of each indicator and these were recorded on a frequency table. No discussions, questions, or comments were allowed until all the participants had given their ratings. At the end of the round-robin, each participant in turn elucidated his/her criteria for the indicator overall rating score. At this stage, open discussions, questions and comments on the indicator were allowed. The discussions were recorded verbatim. The participants were not allowed to revise their individual rating score after the discussion.

**Step 4: Silent generation of new indicators**. After steps 2 and 3 were repeated for all 14 candidate indicators, the participants were given ten minutes to think and write down any missing indicators in line with the central theme question. The new indicator ideas were shared in a round-robin without repeating what had been shared by other participants. These new proposed indicators were written on a flip chart and discussed to ensure all participants understood and approved any new indicator suggestions. The facilitator ensured that all participants were given an opportunity to contribute. From this exercise, new indicators were generated and details defined collectively by the team.

**Step 5: Ranking and sequencing the indicators**. After Step 4, with exclusion of some of the original candidate indicators and addition of new ones based on team discussions, a final list of 15 indicators was generated. Each participant was asked to individually and anonymously rank the final list of the 15 indicators in order of importance, with rank 1 being the most important and rank 15 the least important. The participants were also asked to group the 15 indicators by the implementation priority and sequence into Phase 1 or 2. Phase 1 indicators would be those deemed as not requiring much work to collect, while Phase 2 indicators would require more human input and resources to collect.

## Selection of final indicators

All the individual rankings for each indicator were summed across participants and the final list of prioritized consensus-based EHRs use indicators was derived from the rank order based on the average scores. The ranked indicator list was shared for final discussion and approval by the full team of NGT participants. The relevant indicator reference sheets for every indicator were also updated based on discussions from the NGT exercise. No fixed threshold number was used to select the indicators for inclusion. Finally, the indicator details were reviewed (including indicator definition or how data elements are collected, and indicator calculated) as guided by the NGT session discussions, resulting in the final consensus-based EHRs use reference sheets with details for each indicator.

## Data analysis

Descriptive statistics were computed to investigate statistical differences on the rating of the 14 candidate indicators among the participants. Chi-square test was used to determine if there were statistically significant differences in rating of indicators across each of the SMART dimensions. The ratings totals per SMART dimension from the crosstabs analysis output were summarized in a table (Table 1), indicating the p-value generated from the Chi-square output for each dimension. The variability between the SMART dimensions and the rating was tested using Chi-square since the parameters under investigation were categorical variables (non-parametric data). The totals include rating count and its percentage. Weighted mean for each SMART dimension across all the 14 indicators was calculated to identify how the participants rated various candidate indicators. For the final indicator list, descriptive statistics were computed to determine the average rank score for each indicator and to assign priority numbers from the lowest average score to the highest. As such, the indicator with the lowest average score was considered the most important per the participants' consensus. All analyses were performed in SPSS version 25 (IBM, https://www.ibm.com/analytics/spss-statistics-software). The indicators were also grouped according to implementation phase number assigned by the participants (either 1 or 2) to form the implementation order phases.

## Results

### SMART criteria rating for candidate indicators

The participants rated the collective set of the 14 candidate indicators highly (i.e. 4 or 5) across all the SMART dimensions (Table 1). However, a variation in the totals across the SMART components was due to some participants' non-response in rating some of the components.

From the analysis, the indicators were rated high for specific and time-bound SMART quality dimensions with a mean of 3.96 (p-value = 0.141) for specific and 4.17 (p-value = 0.228) for time-bound. However, the two dimensions did not show any statistically significant difference in how various participants rated them. Measurable, achievable, and relevant dimensions were also high, with the mean of 3.86(p-value = 0.009), 4.01(p-value = 0.039) and 4.27(p-value = 0.023), respectively, and showed statistically significant difference in how the participants rated them across all the indicators.

### Individual indicator ratings

Table 2 shows the participants' overall ratings for each of the 14 candidate indicators on a scale of 1 to 10, reflecting lowest to highest rating respectively. Generally, the participants rated the candidate set of indicators highly with an overall mean rating of 6.6. Data concordance and automatic reports were rated highest with a mean above 8.0. However, the participants rated the observations indicator low with a mean of 3.8, while staff system use, system uptime, and report completeness indicators were moderately rated with a mean of 4.4, 5.9, and 5.8 respectively. The individual indicator ratings and ratings against SMART criteria served as a validation metric for candidate indicators.

### Final indicators list

The NGT team reached a consensus to include all 14 candidate indicators in the final list of indicators, and added one additional indicator, report concordance, for a total of 15 EHRs usage indicators. The final set of indicators fell into four categories, namely (Fig 1 and Table 3):

**Table 1. Summary of the indicators rating on the various SMART quality dimensions.**

| SMART Quality | Responses | | | | | | Total | Mean[b] | P-value |
|---|---|---|---|---|---|---|---|---|---|
| | | Rating[a] of SMART Survey | | | | | | | |
| | | 1 | 2 | 3 | 4 | 5 | | | |
| Specific | Count | 7 | 7 | 18 | 54 | 48 | 134 | 3.96 | 0.141 |
| | Percent | 5.2% | 5.2% | 13.4% | 40.3% | 35.8% | 100.0% | | |
| Measurable | Count | 6 | 12 | 19 | 52 | 43 | 132 | 3.86 | 0.009 |
| | Percent | 4.5% | 9.1% | 14.4% | 39.4% | 32.6% | 100.0% | | |
| Achievable | Count | 4 | 8 | 24 | 42 | 53 | 131 | 4.01 | 0.039 |
| | Percent | 3.1% | 6.1% | 18.3% | 32.1% | 40.5% | 100.0% | | |
| Relevant | Count | 5 | 6 | 11 | 37 | 74 | 133 | 4.27 | 0.023 |
| | Percent | 3.8% | 4.5% | 8.3% | 27.8% | 55.6% | 100.0% | | |
| Time-bound | Count | 5 | 3 | 15 | 51 | 59 | 133 | 4.17 | 0.228 |
| | Percent | 3.8% | 2.3% | 11.3% | 38.3% | 44.4% | 100.0% | | |

[a] Rating Scale 1 = Very Low; 2 = Low; 3 = Neutral; 4 = High; 5 = Very high.

[b] Mean range 1.0–2.5 = Low; 2.6–3.5 = Neutral; 3.6–5.0 = High.

**Table 2. Candidate indicators overall rating.**

| #Indicator | | Indicator overall rating | | | | | | | | | | Total | Mean* |
|---|---|---|---|---|---|---|---|---|---|---|---|---|---|
| | | 1 | 2 | 3 | 4 | 5 | 6 | 7 | 8 | 9 | 10 | | |
| 1 | Data entry statistics | 0 | 0 | 1 | 0 | 1 | 1 | 1 | 2 | 3 | 0 | 9 | 7.1 |
| 2 | Staff system use | 0 | 0 | 3 | 0 | 5 | 1 | 0 | 0 | 0 | 0 | 9 | 4.4 |
| 3 | Observations | 1 | 2 | 2 | 3 | 0 | 0 | 1 | 1 | 0 | 0 | 10 | 3.8 |
| 4 | System uptime | 0 | 0 | 1 | 1 | 2 | 3 | 1 | 1 | 1 | 0 | 10 | 5.9 |
| 5 | Data timeliness | 0 | 0 | 0 | 0 | 0 | 1 | 7 | 2 | 0 | 0 | 10 | 7.1 |
| 6 | Data concordance | 0 | 0 | 0 | 0 | 1 | 0 | 2 | 4 | 1 | 2 | 10 | 8.0 |
| 7 | Data completeness | 0 | 0 | 0 | 0 | 0 | 2 | 3 | 4 | 1 | 0 | 10 | 7.4 |
| 8 | Automatic reports | 0 | 0 | 0 | 1 | 0 | 0 | 0 | 5 | 3 | 1 | 10 | 8.1 |
| 9 | Report timeliness | 1 | 0 | 0 | 0 | 2 | 3 | 0 | 1 | 2 | 1 | 10 | 6.5 |
| 10 | Report completeness | 0 | 0 | 1 | 1 | 1 | 4 | 2 | 1 | 0 | 0 | 10 | 5.8 |
| 11 | Reporting rate | 0 | 0 | 1 | 0 | 1 | 0 | 4 | 1 | 0 | 2 | 9 | 7.1 |
| 12 | Data exchange | 0 | 0 | 0 | 0 | 1 | 2 | 4 | 2 | 0 | 0 | 9 | 6.8 |
| 13 | Standardized terminologies | 0 | 0 | 0 | 1 | 1 | 2 | 3 | 0 | 2 | 0 | 9 | 6.7 |
| 14 | Patient identification | 0 | 0 | 0 | 0 | 0 | 1 | 5 | 2 | 1 | 0 | 9 | 7.3 |
| | Total | 2 | 2 | 9 | 7 | 15 | 20 | 33 | 26 | 14 | 6 | 134 | 6.6 |

* Mean ranges 1.0–4.0 = Low 4.1–6.0 = Neutral 6.1–10.0 = High

1. System Use—these indicators are used to identify how actively the EHRs is being used based on the amount of data, number of staff using the system, and uptime of the system.

2. Data Quality—these indicators are used to highlight proportion and timeliness of relevant clinical data entered into the EHRs. They also capture how well EHRs data captures an accurate clinical picture of the patient.

3. Interoperability—given that a major perceived role of EHRs is to improve sharing of health data, these indicators are used to measure maturity level of implemented EHRs to support interoperability.

**Fig 1. Infographic of key domains for EHRs use indicators.**

**Table 3. The set of validated reporting indicators on EHR system use.**

| # | Domain | Indicator Name | Description | Frequency |
|---|--------|----------------|-------------|-----------|
| 1 | System Use | Data entry statistics | Number and % of patient records entered into system during reporting period | Monthly |
| 2 | System Use | Staff system use | % of providers who entered data into system as expected for at least 90% of encounters | Quarterly |
| 3 | System Use | Observations | Number of observations recorded during period | Quarterly |
| 4 | System Use | System Uptime | % of time system is up when needed during care | Monthly |
| 5 | Data Quality | Clinical data Timeliness | % of clinical provider encounters entered into the EHRs within agreed time period. | Monthly |
| 6 | Data Quality | Variable Concordance | % concordance of data in paper form vs data in EHRs | Quarterly |
| 7 | Data Quality | Variable Completeness | % of required data elements contained in EHRs | Quarterly |
| 8 | Interoperability | Data Exchange | Automatic exchanging of data with different systems | Quarterly |
| 9 | Interoperability | Standardized Terminologies | % of terms that are mapped to standardized terminologies or national dictionary. | Yearly |
| 10 | Interoperability | Patient identification | % of nationally accepted patient identification instances in the EHRs. | Quarterly |
| 11 | Reporting | Automatic Reports | Proportion of expected reports generated automatically by system | In-line with PEPFAR[a] reports |
| 12 | Reporting | Reporting Rate | Proportion of expected reports that are actually submitted | Monthly |
| 13 | Reporting | Report Timeliness | Timeliness of expected reports to national reporting system | Monthly |
| 14 | Reporting | Report Completeness | Completeness of expected reports to national reporting system | In-line with PEPFAR reports |
| 15 | Reporting | Report Concordance | % concordance of data contained in paper-derived reports compare to report data derived from the EHRs | Biannual |

[a] Monitoring, Evaluation, and Reporting [MER] indicators reporting by PEPFAR initiated HIV programs

**Table 4. Ranking of finalized EHRs use indicators.**

| Indicator Ranking | Indicator Name | Average Score Mean (SD) |
|---|---|---|
| 1 | Data Entry Statistics | 2.78 (2.33) |
| 2 | System Uptime | 4.56 (5.22) |
| 3 | EHR Variable concordance | 6.44 (2.80) |
| 4 | EHR Variable Completeness | 6.56 (3.32) |
| 5 | Report Concordance | 6.67 (4.66) |
| 6 | Staff system use | 6.78 (4.64) |
| 7 | Clinical Data Timeliness | 7.33 (4.61) |
| 8 | Report Completeness | 7.89 (2.98) |
| 9 | Patient Identification | 8.00 (4.33) |
| 10 | Data exchange | 8.67 (4.12) |
| 11 | Reporting timeliness | 9.00 (3.61) |
| 12 | Automatic Reports | 10.33 (2.83) |
| 13 | Observations | 11.56 (4.12) |
| 14 | Standardized Terminologies | 11.56 (2.87) |
| 15 | Reporting Rate | 11.89 (2.76) |

4. Reporting—aggregation and submission of reports is a major goal of the implemented EHRs, and these indicators capture how well the EHRs are actively used to support the various reporting needs.

As part of the NGT exercise, the details of each of the indicators was also refined. S3 Appendix presents the detailed EHRs MER document, with agreed details for each indicator provided. In this document, we also highlight the changes that were suggested for each indicator as part of the NGT discussions.

### Indicator ranking

The score and rank procedure generated a prioritized consensus-based list of EHRs use indicators with a score of 1 (highest rated) to 15 (lowest rated). As such, a low average score Mean' meant that the particular indicator was on average rated higher by the NGT participants. Table 4 presents the ordered list of ranking for the indicators as rated by nine of the NGT participants as one participant was absent during this NGT activity. Data Entry Statistics and System Uptime indicators were considered to be the most relevant in determining EHRs usage, while Reporting Rate indicator was rated as least relevant.

### Indicator implementation sequence

Nine of the 15 indicators were recommended for implementation in the first phase of the indicator tool rollout, while the other six indicators were recommended for Phase 2 rollout (Table 5). The implementation sequence largely aligns with the indicator priority ranking by the participants (Table 4). The indicators proposed for Phase 1 implementation are a blend from the four indicator categories but are mostly dominated by the System Use subcategory.

### Discussion

To the best of our knowledge, this is the first set of systematically developed indicators to evaluate the actual status of EHRs usage once an implementation is in place within LMIC settings.

**Table 5. Recommended implementation sequence of the EHRs use indicators.**

| Implementation sequence | Indicator name | |
| --- | --- | --- |
| | Phase 1 | Phase 2 |
| 1 | Data Entry Statistics | Standardized Terminologies |
| 2 | System Uptime | Observations |
| 3 | EHRs data concordance | Automatic Reports |
| 4 | EHRs Data Completeness | Report timeliness |
| 5 | Staff system use | Reporting Rates |
| 6 | Clinical Data Timeliness | Data Exchange |
| 7 | Report Concordance | |
| 8 | Reporting Completeness | |
| 9 | Patient Identification | |

At the completion of the modified NGT process, we identified 15 potential indicators for monitoring and evaluating status of actual EHRs use. These indicators take into consideration constraints within the LMIC's setting such as system availability, human resource constraints, and infrastructure needs. Ideally, an IS implementation is considered successful if the system is available to the users whenever and wherever it is needed for use [46]. Clear measures of system availability, use, data quality, and reporting capabilities will ensure that decision makers have clear and early visibility into success and challenges facing system use. Further, the developed indicators allow for aggregation of usage indicators to evaluate performance of systems by type, regions, facility level, and implementing partners.

An important consideration of these indicators is the source of measure data. Most published studies on evaluating success of information system focus on IS use indicators or variables such as ease of use, frequency of use, extent of use, and ease of learning, mostly evaluated by means of self-reporting tools (questionnaires and interviews) [19,39,47]. As such, the resulting data can be subjective and prone to bias. We tailored our indicators to ensure that most can be computer-generated through queries, hence incorporating objectivity into the measurement. However, a few of these indicators, such as data entry statistics as well as those on concordance (variable concordance and report concordance) derive measure data from facility records in addition to computer logs data.

Although the NGT expert panel was national, we are convinced the emerging results are of global interest. First, we developed the indicators in-line with the internationally renowned PEPFAR Monitoring, Evaluation, and Reporting (MER) indicators Reference Guide [35]. Secondly, the development process was mainly based on methodological criteria that are valid everywhere [48,49]. Furthermore, the indicators are not system-specific and hence can be used to evaluate usage of other types of EHRs, including other clinical information systems implementations like laboratory, radiology, and pharmacy systems. However, we recognize that differences exist in systems database structure; hence, the queries to determine indicator measures data from within each system will need to be customized and system-specific. It is important to also point out that these indicators are not based on real-time measures and can be applied both for point of care and non–point of care systems.

The selected set of indicators have a high potential to determine the status of EHRs implementations considering that the study participants rated all five SMART dimensions high (over 70%) across all the indicators. Further, the indicators reference guide provides details on "how to collect" and the sources of measure data for each indicator (S3 Appendix). This diminishes the level of ambiguity in regard to measurability of the indicators. Nonetheless, some of the indicators need countries to define their own thresholds and reporting frequencies. For

instance, a country would need to define the length of acceptable time duration within which a clinical encounter should be entered into the EHRs for that encounter to be considered as having been entered in a timely fashion. As such, the indicator and reference guide need to be adapted for specific country and use context. Despite staff system use and observations indicators low overall rating (4.4 and 3.8 respectively), they were included in the final list of indicators after consensus-based discussions as part of the NGT exercise. We believe this is due to the indicators' direct role in determining system usage and the fact that they were scored highly in the SMART assessment. Further assessment with a wider group of intermediate system users would be beneficial to estimate the value of the indicators in question before rendering them irrelevant.

This study has several limitations. It was based on a multidisciplinary panel of 10 experts, which is adequate for most NGT exercises, but still has a limited number of individuals who might not reflect all perspectives. On average, 5–15 participants per group are recommended depending on the nature of the study [50,51]. The low ranking of Data Exchange and Standardized Terminologies indicators indicate that the participants might have limited knowledge or appreciation of certain domains and their role in enhancing system use. Further, all participants were drawn from one country. Nevertheless, a notable strength was the incorporation of participants from more than one EHRs (KenyaEMR and IQCare systems) and a diverse set of expertise. In addition, the derived indicators do not assess the "satisfaction of use" dimension outlined in Delone & McLean mode [39] and future work should extend the indicators to explore this dimension.

A next step in our research is to conduct an evaluation on actual system use status for an information system rolled-out nationally, using the developed set of indicators. We will also evaluate the real-world challenges of implementing the indicators and refine them based on the findings. We also anticipate sharing these indicators with a global audience for input, validation, and evaluation. We are cognizant of the fact that the indicators and reference guides are living documents and they are bound to evolve over time, given the changing nature of the IS field and maturity of EHRs implementations.

## Conclusion

An NGT approach was used to generate and prioritize a list of consensus-based indicators to assess actual EHRs usage status in Kenya. However, the indicators can be applicable to LMICs and similar contexts. This list of indicators can allow for monitoring and aggregation of EHRs usage measures to ensure that appropriate and timely actions are taken at institutional, regional, and national levels to assure effective use of EHRs implementations.

## Supporting information

**S1 Appendix. System usage indicator template.**
(PDF)

**S2 Appendix. Indicator rating form.**
(PDF)

**S3 Appendix. Monitoring, Evaluation and Reporting (MER v1.0): Electronic Health Record (EHR) system usage indicator reference guide.**
(DOCX)

**S1 File.**
(XLSX)

## Acknowledgments

Authors would like to acknowledge the US Centers for Disease Control and Prevention (CDC) for providing input into the candidate set of indicators. We also appreciate the insights and contributions from all the workshop participants drawn from CDC-Kenya, Kenyan Ministry of Health, Palladium (EHRs development partners), EHRs implementing partners, Moi University, and EHRs users.

## Author Contributions

**Conceptualization:** Philomena Ngugi, Xenophon Santas, Martin C. Were.

**Data curation:** Philomena Ngugi.

**Formal analysis:** Philomena Ngugi, Ankica Babic, Martin C. Were.

**Funding acquisition:** Martin C. Were.

**Investigation:** Philomena Ngugi, James Kariuki, Martin C. Were.

**Methodology:** Philomena Ngugi, Violet Naanyu, Martin C. Were.

**Project administration:** Philomena Ngugi, Ankica Babic, Martin C. Were.

**Resources:** Philomena Ngugi, Ankica Babic, Violet Naanyu, Martin C. Were.

**Software:** Philomena Ngugi.

**Supervision:** Ankica Babic, Martin C. Were.

**Validation:** Philomena Ngugi, Ankica Babic, James Kariuki, Martin C. Were.

**Visualization:** Philomena Ngugi.

**Writing – original draft:** Philomena Ngugi, Ankica Babic, Martin C. Were.

**Writing – review & editing:** Philomena Ngugi, Ankica Babic, James Kariuki, Xenophon Santas, Violet Naanyu, Martin C. Were.

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
