## [Decision Letter · Decision Letter 0]

10 Dec 2020

PONE-D-20-26805

Development of standard indicators to assess use of electronic health record systems implemented in low-and medium-income countries

PLOS ONE

Dear Dr. Ngugi,

Thank you for submitting your manuscript to PLOS ONE. After careful consideration, we feel that it has merit but does not fully meet PLOS ONE’s publication criteria as it currently stands. Therefore, we invite you to submit a revised version of the manuscript that addresses the points raised during the review process.

We look forward to receiving your revised manuscript.

Kind regards,

Chaisiri Angkurawaranon

Academic Editor

PLOS ONE

Journal Requirements:

2.) Thank you for including your ethics statement:  "The study obtained written approval by the Institutional Review and Ethics Committee at Moi University, Eldoret (MU/MTRH-IREC approval Number FAN:0003348).".   

Please provide additional details regarding participant consent. In the ethics statement in the Methods and online submission information, please ensure that you have specified what type you obtained (for instance, written or verbal, and if verbal, how it was documented and witnessed). If your study included minors, state whether you obtained consent from parents or guardians. If the need for consent was waived by the ethics committee, please include this information.

3.) We note that you have indicated that data from this study are available upon request. PLOS only allows data to be available upon request if there are legal or ethical restrictions on sharing data publicly. For more information on unacceptable data access restrictions, please see http://journals.plos.org/plosone/s/data-availability#loc-unacceptable-data-access-restrictions.

Reviewers' comments:

Reviewer's Responses to Questions

**Comments to the Author**

1. Is the manuscript technically sound, and do the data support the conclusions?

Reviewer #1: Yes

Reviewer #2: Yes

2. Has the statistical analysis been performed appropriately and rigorously? 

Reviewer #1: Yes

Reviewer #2: Yes

3. Have the authors made all data underlying the findings in their manuscript fully available?

Reviewer #1: Yes

Reviewer #2: Yes

4. Is the manuscript presented in an intelligible fashion and written in standard English?

Reviewer #1: Yes

Reviewer #2: Yes

5. Review Comments to the Author

Reviewer #1: Thank you for the opportunity to review this manuscript. In this manuscript the authors describe a study to develop and validate a set of electronic health record systems use indicators for LMICs. This is a well-constructed and reported study with sound methodology. The reported consensus-based set of indicators will be of interest to a broad, international readership. Further, the manuscript is well supported by the provision of a comprehensive indicator reference guide in the supplementary material.

Reviewer #2: This manuscript addresses an interesting topic, the evaluation of the implementation of EHRs in Low or Midlle-income countries (LMICs). It relies on a consensus method to define 15 indicators and a timeline to use them.

This paper seems to me to ask the following questions:

- The choice of the 14 initial indicators seems to be inspired by the well-known analytical framework of De Loine & Mac Lean. One point of this model is to emphasize the "satisfaction of use" dimension. However, this does not appear in the article, the ‘use‘ and ‘ease of use’ dimension being the only one mentioned. This is a problem when you consider that part of the issues lies in user acceptability, regardless of the architectural and technical quality of the EHR. This is emphasized in the introduction. Authors should position themselves more clearly on this point in the introduction and discussion parts.

- The description of the participants is a key point in these consensus methods. The number of participants is quite low, coming from a single country if we refer to what is specified in the limit part. It is mentioned they have various expertise in this part. The description of the participants and their profile should be further developed in the method section.

- The interest of the analysis of variability in rating based on chi-square test is not explained.

- The current development context of EHR in these countries is not sufficiently described. It would be important to specify whether the EHR has been already developed, in part or completely, its objectives of use (accountability, coordination between professionals), and its relationship with a paper sheet record. Understanding the maturity of EHRs’ implementation remains a key point, several countries facing this question until recently (2012. Couralet M, et al. Method for developing national quality indicators based on manual data extraction from medical records, BMJ Quality and Safety. 22-2:155-162).

- Two indicators are rated very low, but are selected, which suggests that the rating did not have influence. It is important to justify why they were selected.

Staff system use 0 0 3 0 5 1 0 0 0 0 9 4.4

Observations 1 2 2 3 0 0 1 1 0 0 10 3.8

These questions deserve to be addressed to improve the manuscript.

6. PLOS authors have the option to publish the peer review history of their article (what does this mean?). If published, this will include your full peer review and any attached files.

Reviewer #1: No

Reviewer #2: No

---

## [Author Response · Author response to Decision Letter 0]

17 Dec 2020

Re: Manuscript PONE-D-20-26805: Development of standard indicators to assess use of electronic health record systems implemented in low-and medium-income countries.

We appreciate the review by PLOS ONE Journal Scientific Program Committee of our Manuscript entitled “Development of standard indicators to assess use of electronic health record systems implemented in low-and medium-income countries” and are grateful for the opportunity to respond comprehensively to the reviewers’ comments.

Please find our responses to all the comments by the reviewers below:

Reviewers’ comments

Reviewer 1:

Thank you for the opportunity to review this manuscript. In this manuscript the authors describe a study to develop and validate a set of electronic health record systems use indicators for LMICs. This is a well-constructed and reported study with sound methodology. The reported consensus-based set of indicators will be of interest to a broad, international readership. Further, the manuscript is well supported by the provision of a comprehensive indicator reference guide in the supplementary material.

We appreciate the positive comments on our work and manuscript. 

Reviewer 2: 

This manuscript addresses an interesting topic, the evaluation of the implementation of EHRs in Low or Middle-income countries (LMICs). It relies on a consensus method to define 15 indicators and a timeline to use them.

This paper seems to me to ask the following questions:

1. The choice of the 14 initial indicators seems to be inspired by the well-known analytical framework of De Loine & Mac Lean. One point of this model is to emphasize the "satisfaction of use" dimension. However, this does not appear in the article, the ‘use‘ and ‘ease of use’ dimension being the only one mentioned. This is a problem when you consider that part of the issues lies in user acceptability, regardless of the architectural and technical quality of the EHR. This is emphasized in the introduction. Authors should position themselves more clearly on this point in the introduction and discussion parts.

We do appreciate reviewer pointing out the need to include “satisfaction of use” dimension which is one of the constructs of D&M IS success model. Satisfaction of use is an important dimension to evaluate, and we do agree that systems should be designed to allow for feedback on user satisfaction. However, this component was outside the scope of this project, which focussed on actual system use components. In response to the reviewer’s comment, we have added the following sentence in the Discussion:

 “In addition, the derived indicators do not assess the "satisfaction of use" dimension outlined in Delone & McLean model,[39] and future work should extend the indicators to explore this dimension.” – Line 360-362.

2. The description of the participants is a key point in these consensus methods. The number of participants is quite low, coming from a single country if we refer to what is specified in the limit part. It is mentioned they have various expertise in this part. The description of the participants and their profile should be further developed in the method section.

We have taken note of the comments. We have added further description of the participants as follows: 

The NGT participants’ average age was 40 years where majority were males (69%). The participants included: the researchers acting as facilitators; one qualitative researcher (an associate professor and lecturer); two MoH representatives from the Division of Health Informatics and M&E (health information systems management experts); one Service Development Partners (SDPs) representative (oversees EHRs implementations and training of users); Four users of the EHRs (clinical officers (2) & Health records information officers (2)); CDC funding agency representative (an informatics service fellow in the Health Information Systems); and two representatives from the EHRs development and implementing partners (Palladium and International Training and Education Center for Health (I-TECH)), who have been involved in the EHRs implementations and who selected sites for EHRs implementations

The manuscript was updated accordingly (starting at line 145-155).

3. The interest of the analysis of variability in rating based on chi-square test is not explained.

We appreciate and agree with the reviewer’s comment. We have added an explanation on this as follows:

“The variability between the SMART dimensions and the rating was tested using Chi-square since the parameters under investigation were categorical variables (non-parametric data).”

The manuscript was updated accordingly (starting at line 226-227)

4. The current development context of EHR in these countries is not sufficiently described. It would be important to specify whether the EHR has been already developed, in part or completely, its objectives of use (accountability, coordination between professionals), and its relationship with a paper sheet record. Understanding the maturity of EHRs’ implementation remains a key point, several countries facing this question until recently (2012. Couralet M, et al. Method for developing national quality indicators based on manual data extraction from medical records, BMJ Quality and Safety. 22-2:155-162).

We do appreciate reviewers pointing out the need to add more literature on EHRs in the context of our study. In the introduction part of the manuscript, we have mentioned that implementations of EHRs exist and have continued to grow, and the driving factors, but their success has not been investigated. We have, however, added an example of implementations in Kenya as follows:

“For example, Kenya has had over 1000 electronic medical systems (EMRs) implementations progressively since 2012 in both private and public facilities supporting patient data management mainly in HIV care and treatment.” (Introduction, paragraph 1, Line 52)

5. Two indicators are rated very low, but are selected, which suggests that the rating did not have influence. It is important to justify why they were selected.

Staff system use 0 0 3 0 5 1 0 0 0 0 9 4.4

Observations 1 2 2 3 0 0 1 1 0 0 10 3.8

These questions deserve to be addressed to improve the manuscript.

We have taken note of the comments. We have added an explanation on this as follows:

“Despite staff system use and observations indicators low overall rating (4.4 and 3.8 respectively), they were included in the final list of indicators after consensus-based discussions as part of the NGT exercise. We believe this is due to the indicators’ direct role in determining system usage and the fact that they were scored highly in the SMART assessment. Further assessment with a wider group of intermediate system users would be beneficial to estimate the value of the indicators in question before rendering them irrelevant.” 

We have updated the manuscript accordingly (starting at line 346-351)

Editors comments:

We have followed the provided guidelines and conformed to manuscript style requirements

2. Thank you for including your ethics statement: "The study obtained written approval by the Institutional Review and Ethics Committee at Moi University, Eldoret (MU/MTRH-IREC approval Number FAN:0003348).". 

Please provide additional details regarding participant consent. In the ethics statement in the Methods and online submission information, please ensure that you have specified what type you obtained (for instance, written or verbal, and if verbal, how it was documented and witnessed). If your study included minors, state whether you obtained consent from parents or guardians. If the need for consent was waived by the ethics committee, please include this information.

We have taken note of the comments. We have amended the ethics statement in the methods section of the manuscript as follows:

“All participants filled a printed consent form before taking part in the study”. (starting at line 156-157)

We also added the text in the Ethics statement field of the submission form.

If there are ethical or legal restrictions on sharing a de-identified data set, please explain them in detail (e.g., data contain potentially sensitive information, data are owned by a third-party organization, etc.) and who has imposed them (e.g., an ethics committee). Please also provide contact information for a data access committee, ethics committee, or other institutional body to which data requests may be sent. 

If there are no restrictions, please upload the minimal anonymized data set necessary to replicate your study findings as either Supporting Information files or to a stable, public repository and provide us with the relevant URLs, DOIs, or accession numbers. For a list of acceptable repositories, please see http://journals.plos.org/plosone/s/data-availability#loc-recommended-repositories.

We appreciate your clarification on data availability. We have revised the cover letter to include this as guided. We have also uploaded the study data as supporting information (S4_File.xlsx).

Thank you once again for considering our manuscript in PLOS ONE.

Sincerely,

Philomena Ngugi

waruharip@gmail.com

Corresponding author

---

## [Editor Report · Decision Letter 1]

21 Dec 2020

Development of standard indicators to assess use of electronic health record systems implemented in low-and medium-income countries

PONE-D-20-26805R1

Dear Dr. Ngugi,

We’re pleased to inform you that your manuscript has been judged scientifically suitable for publication and will be formally accepted for publication once it meets all outstanding technical requirements.

Kind regards,

Chaisiri Angkurawaranon

Academic Editor

PLOS ONE
---

## [Editor Report · Acceptance letter]

23 Dec 2020

PONE-D-20-26805R1 

Development of standard indicators to assess use of electronic health record systems implemented in low-and medium-income countries 

Dear Dr. Ngugi:

I'm pleased to inform you that your manuscript has been deemed suitable for publication in PLOS ONE. Congratulations! Your manuscript is now with our production department. 

Kind regards, 

on behalf of

Dr. Chaisiri Angkurawaranon 

Academic Editor

PLOS ONE